**communications** sustainability

# Large language model reveals an increase in climate contrarian speech in the United States Congress
Travis G. Coan [1] ✉, Ranadheer Malla[1], Mirjam O. Nanko [1], William Kattrup [2], J. Timmons Roberts[2], John Cook[3] & Constantine Boussalis[4]

Contrarian voices on climate change have reached the highest levels of government in the United States: the US Congress has been a focal point for climate obstruction, playing an important role in impeding US climate action and stymieing global negotiations. We take a closer look at these voices by examining Congressional speeches from 1994 to 2023, a period containing important moments in the history of climate (in)action in the US. We contribute to the literature by 1) developing a scalable large language model (LLM) to accurately classify climate contrarianism in Congressional speech and 2) offering a systematic analysis of the specific contrarian claims in Congress. We demonstrate that an increasing proportion of speeches are critiquing the costs of climate solutions and Republicans are roughly 12 times more likely than Democrats to make contrarian claims. Statistical analysis further suggests that demographic factors and district-level fossil fuel employment predict claims making in floor speeches.

An established body of scholarship examines the role of an organized and well-funded climate change "countermovement" in manufacturing doubt on climate science and promoting discourses of delay on climate policy[1–4]. A significant—and growing—number of studies highlight the deleterious effect of misinformation on healthy climate debate, demonstrating the ways in which misinformation 1) negates accurate information[5,6] and reduces climate literacy[7], 2) impedes decision making and evidence-based public policy[8], 3) increases political polarization[9], and 4) erodes public trust in science[4,10]. And while much of the literature centers on climate contrarianism in the United States, studies suggest that the arguments and tactics employed by the countermovement are spreading beyond their largely North American origins[11]. Evidence on the transnational diffusion of misinformation to key developing countries further underscores the importance of developing approaches to detect, monitor, and debunk false and misleading claims on climate change and its solutions[12,13].

Contrarian arguments appear to influence the climate policy process. In the US, contrarian voices permeate the highest levels of government: the US Congress has long served as a bastion of climate skepticism and played a prominent role in obstructing climate action[14,15]. For instance, one recent study documents the presence of both overt "climate change denial" (i.e., the explicit questioning of climate science) and "anecdotal denial" (i.e., deploying analogies and storytelling to subtly question the severity of climate change)[16]. However, to our knowledge, there is no research on the prevalence of specific contrarian claims in Congress and their varying distributions across factors such as party, state, time, and external funding. This remains particularly important to study, not just to understand the influence of obstructionist discourse on policy outcomes, but because evidence suggests that Congressional dynamics and elite cues are one of the most important drivers in public opinion on climate change[17,18].

Whether analyzing contrarian discourse in the entire Congressional record or across modern online media, one thing is clear: the volume of content is too vast to be read and analyzed manually. Increasingly, researchers are turning to computational approaches to augment and extend manual content analyses. Machine learning approaches have been used to understand contrarian discourse in a range of sources, including conservative think tank websites, climate skeptic blogs, Congressional floor speeches, online news media, and social media. While early work focused on unsupervised machine learning approaches to examine the *general* topics or themes promoted by conservative think tanks (CTTs) in the US[19,20], more recent scholarship has developed supervised machine learning models to detect *specific* contrarian claims[21,22]. The move from general to specific computational models has important implications, ensuring that model-based predictions are at a level of detail appropriate for "technocognitive" approaches that integrate psychological research with automated detection,

[1]Centre for Climate Communication and Data Science, University of Exeter, Exeter, UK. [2]Brown University, Climate and Development Lab, Providence, RI, USA. [3]Melbourne Centre for Behaviour Change, University of Melbourne, Parkville, VIC, Australia. [4]Department of Political Science, Trinity College Dublin, Dublin 2, Ireland. ✉e-mail: t.coan@exeter.ac.uk

fact-checking algorithms, and digital intervention platforms to combat misinformation[23].

Our work builds on and extends recent machine learning work on detecting specific contrarian claims. First, we modify and extend Coan et al.'s CARDS taxonomy[21] to better represent solutions-focused skepticism and clarify the connection between the *claims* outlined in CARDS and the *discourses* of delay outlined in Lamb et al.'s research[3]. Next, we develop a flexible, fine-tuned large language model (LLM) that offers several key improvements over earlier models (see *Methods*). Lastly, we utilize our model to offer the first comprehensive history of contrarian claims in US congressional floor speeches and examine the correlates of claims-making by integrating information on Members of Congress (MoCs) and their districts. We find that contrarian claims spike during major climate policy moments (1997, 2008-9, and 2015), with "solutions increase costs" comprising roughly one-third of all contrarian claims made. For every one speech that included a contrarian claim from a Democrat, there were roughly 13 speeches that included a claim from Republicans. Further, more conservative Republicans, younger members, and those representing constituencies with higher fossil fuel employment or receiving more fossil fuel campaign contributions are also more likely to challenge climate science or policy on the floor of Congress. Our results show that political party affiliation and ideology are by far the strongest predictors of contrarian speech, even after controlling for demographic and economic factors.

## Results

### Taxonomy development
Our work builds on and extends the comprehensive taxonomy of contrarian claims introduced in Coan et al.[21]. Often referred to as the CARDS (**C**omputer-**A**ssisted **R**ecognition of climate change **D**enial & **S**kepticism) taxonomy, Coan et al. provide a hierarchical structure for organizing claims into five major level-1 claims: 1) it's not happening, 2) it's not us, 3) the impacts won't be bad, 4) climate solutions won't work, and 5) climate science and proponents of climate action are biased. Each major claim, moreover, was accompanied by more detailed level-2 and level-3 claims, thereby offering a comprehensive framework for organizing claims which is granular enough to facilitate debunking and inoculation efforts[6,9].

The original CARDS taxonomy, however, had several limitations in terms of comprehensiveness and coherence. These limitations were especially acute for the climate "solutions won't work" category, which has been growing in prevalence and is particularly relevant to Congressional speech. This level-1 claim was the least refined among the major claims, primarily because the original taxonomy was developed with a focus on online publications from conservative think tanks and contrarian blogs, with no specific emphasis on policy-related nuances. Further, as Coan et al. describe, the original taxonomy included several mainstream claims (e.g., claims supporting the use of nuclear energy or that CCS technology is unproven) which appeared in their sample of documents, causing some confusion between claims that are contrarian and claims that contrarians tend to make. The challenges introduced by these limitations were exemplified by the comparatively lower performance of the trained classifier on some of the level-2 claims in the "solutions won't work" category reported in Coan et al.[21].

To address these limitations, we developed the revised taxonomy in Fig. 1. Specifically, this revision 1) re-organizes and extends the "solutions won't work" category, separating attacks on climate solutions from claims supporting the continued use of fossil fuels and 2) separates the original "climate science is unreliable" into claims that attack climate science, data, and processes from those that attack climate *scientists* and other proponents of climate action. Section S.1 in the Supplementary Information provides a detailed overview and justification of the proposed changes to the original CARDS taxonomy (see Table S1 in the Supplementary Information for a complete list of claims). Table 1 further illustrates the connections between the general discursive strategies outlined in Lamb et al.[3] and the specific claims outlined in Fig. 1. As demonstrated in the table, the revised taxonomy includes claims closely related to redirecting responsibility, pushing for non-

transformative solutions, emphasizing the downsides of proposed action, and surrendering to effects of climate change.

It is important to reiterate that not all claims in Fig. 1 are examples of climate misinformation, and any model developed to classify these claims cannot, on its own, detect misinformation. While the systematic deconstruction[24] of these claims has shown that almost all contain reasoning fallacies[25], the context in which a claim is made plays an important role in determining whether a claim is false or misleading. For instance, there is a significant difference between the claim that "climate models are uncertain" (which they are) and "climate models are uncertain, therefore we shouldn't trust climate science" (which is misleading). Focusing only on *explicit* forms of climate misinformation, however, would miss a substantial proportion of the problematic messages spread, particularly on prominent social media platforms where claims are often implicit and may draw on coded language and euphemisms. Conspiracy theories and ad hominem attacks designed to undermine the credibility of climate science, while comprising 60% of climate misinformation on Twitter/X[22], often employ non-empirical arguments, making them difficult to fact-check. Arguments can be misleading even through the use of factual statements if they withhold the broader context in order to portray a misleading expression—a technique known as paltering[26]. Also challenging are arguments that rely on hidden premises or unstated assumptions, which by not being explicitly stated can avoid being held to account by fact-checking[27]. This is especially true for climate misinformation, with over 90% of the claims in the CARDS taxonomy containing hidden premises that committed reasoning fallacies[25]. The limitations of fact-checking to address all forms of climate misinformation underscore the importance of complementing fact-based approaches with logic-based methods that scrutinize rhetorical techniques and reasoning fallacies in contrarian claims. Although several recent studies have made progress on generative approaches to debunking climate misinformation[28], in particular in detecting reasoning fallacies in climate misinformation[29], the model developed below *assists* but does not replace fact-checking and systematic debunking efforts.

### A new model for classifying contrarian claims
As detailed in Methods, we develop a new model for multi-label classification of the *specific* claims outlined in Fig. 1. We introduce a flexible and efficient procedure—reverse engineered chain-of-thought (RECoT) fine-tuning—to offer a scalable LLM with competitive levels of performance when compared to larger frontier models. Our best performing fine-tuned model (CARDS-mini-Sonnet-2024-12-05) achieved an F1-score of 0.852 (precision = 0.866, recall = 0.848) when classifying at level 3 of the taxonomy, which closely approximates those of Claude-Sonnet-3.7 (F1 = 0.881, precision = 0.890, recall = 0.879, accuracy = 0.836), the largest, highest-performing zero-shot model at the time of this analysis (see Methods for details). RECoT fine-tuning delivered substantial performance improvements over the base GPT-4o-Mini model (zero-shot), with increases of 34.6 percentage points in F1-score. Notably, these findings have practical benefits for researchers in the field, as our fine-tuned model achieves competitive performance while being ~10 times more cost-effective than Claude-Sonnet-3.7 for inference, making it viable for large-scale text classification tasks. Given its strong performance and scalability, we utilize the CARDS-mini-Sonnet-2024-12-05 to generate the data used for the analysis outlined in subsequent sections.

### Prevalence of detailed contrarian claims in congressional testimonies
Figure 2 provides an overview of contrarian claims making in congressional floor speeches over the period from 1994 to 2024. Unsurprisingly, as demonstrated in Fig. 2a, b, claims on the harmful effects of climate solutions and advocating for the necessity and benefits of fossil fuels dominate the discourse over this three decade period. First, arguments focusing on the cost of proposed climate action ("Solutions increase costs") are far and away the most dominant theme, representing over one-third of all contrarian

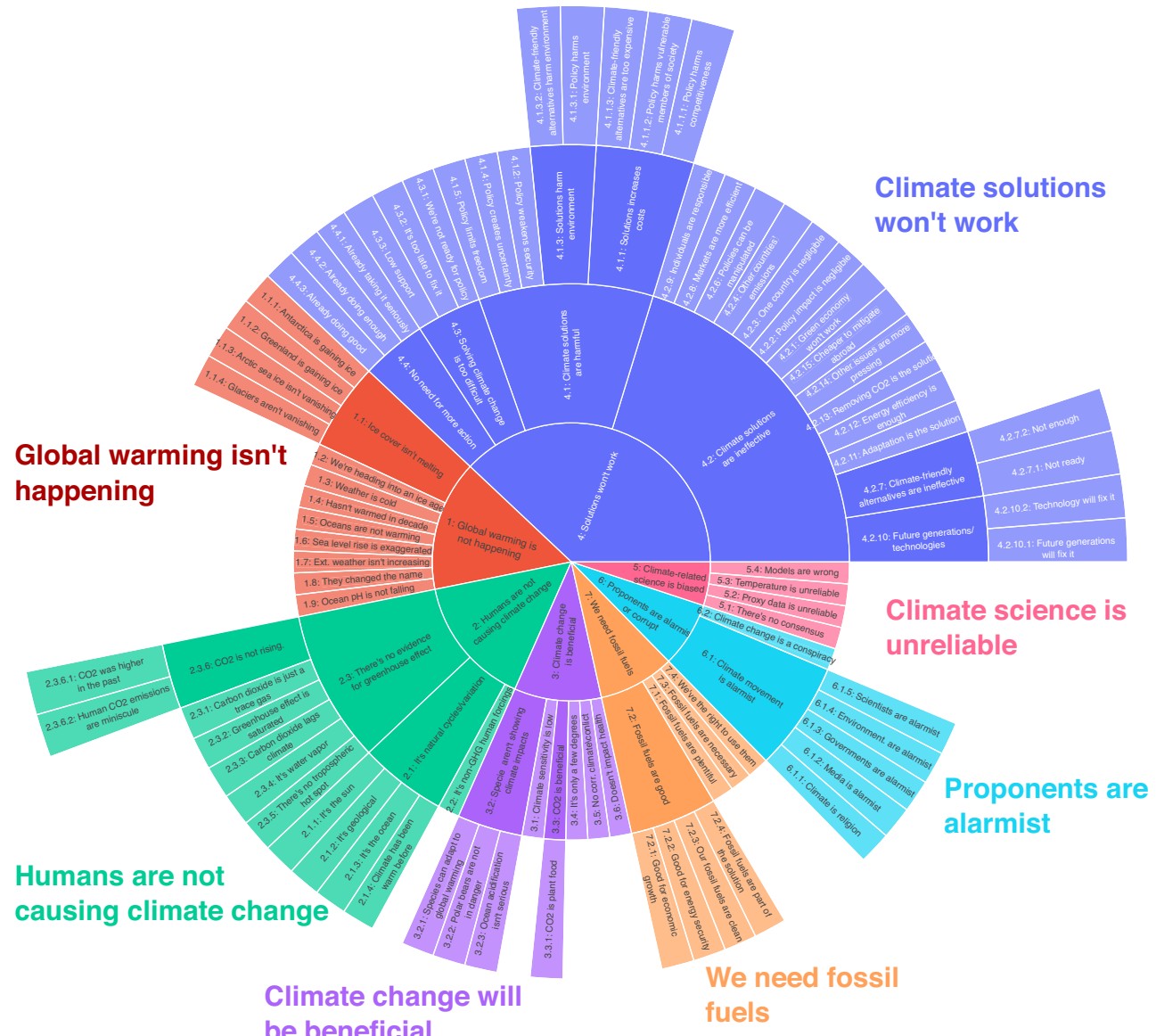

**Fig. 1 | Taxonomy of contrarian claims.** This figure shows the revised version of the CARDS[21] taxonomy of contrarian claims. The revised taxonomy makes substantial changes to category 4 ("Solutions won't work") and category 5 ("Science is unreliable") of the original CARDS taxonomy (see Section S.1 of the Supplementary Information for details of these changes).

claims made. An examination of the detailed claims used to describe the economic harms of climate policy suggests that Members of Congress (MoCs) are slightly more likely to focus on the harms to individuals (e.g., policies kill jobs or hurt vulnerable members of society) rather than larger concerns about the harms to economic competitiveness. Claims that climate policy limits freedom and leads to regulatory overreach are also quite common, with these speeches often employing rhetorical strategies to suggest that policy solutions represent a "war on American energy." Second, claims stressing the benefits or necessity of fossil fuels also appear frequently, often made in the context of the benefits of the continued exploration and use of fossil fuels for energy dependence and economic growth. Lastly, though much less frequent, MoCs have used floor speeches as a platform for questioning climate science, mainly by casting doubt on the scientific consensus on climate change or arguing that climate change is the result of natural variability.

Turning next to Fig. 2c through e, we find that claims making in congressional speech varies considerably over time and often in predictable

ways. During the early Kyoto Protocol period (1995–2005), contrarian claims remained relatively low, although we observe a significant spike in speech questioning the scientific consensus and attacking the reliability of climate science in 1997 (i.e., the year the Kyoto Protocol was negotiated). Yet, it was not until the run-up to the American Clean Energy and Security Act (ACES) in 2009 (i.e., the Waxman-Markey cap-and-trade bill) that contrarian claims started to appear frequently in floor speeches, showing up in roughly 15% of all speeches mentioning climate change. Triggered in large part by rising gasoline prices throughout the cap-and-trade period (2005–2010), statements on the benefits and necessity of fossil fuels reached an all time high in 2008, typified by chants of "drill baby drill" at the 2008 GOP national convention, which framed more drilling as a patriotic fix to pain at the pump. By 2009, the conversation had shifted to challenging climate solutions, while we also observe distinct spikes in claims of climate alarmism, challenges to climate science, arguments questioning whether climate change is happening, and whether it is caused by human activity. In the period following the failed effort to pass cap-and-trade legislation,

contrarian claims making continues to closely align with domestic and international policy events. For instance, many of the time series rise through 2015 (a year which included the roll-out of Obama's Clean Power Plan and the Paris Agreement), dip slightly in the early days of the Trump presidency, and then rise again as Democrats—led by the Biden administration—looked to revive big-ticket climate legislation. These data suggest that floor speeches are roughly five times more likely to challenge climate solutions during the Biden period (from 2021) than during the mid-2000

period. The shift from denial rhetoric to discourses of delay has been well documented by scholars and journalists. Our analysis demonstrates that this shift occurred in Congress, too. Importantly, this shift is not characterized by a sharp decline in denial rhetoric, but rather by a dramatic increase in delay arguments (which appear in our taxonomy as "solutions won't work" and "fossil fuels are good"). In other words, denial rhetoric continues to appear in climate discourse in Congress, but there has been a strong addition of discourses of delay.

Figure 3 reveals stark partisan differences in the prevalence and geographic intensity of climate contrarian claims across Congressional speeches from 1994 to 2023. The horizontal bar chart demonstrates that Republicans vastly outnumber Democrats in making contrarian claims across all categories, with particularly pronounced disparities in the two most frequent claim types: "Solutions Won't Work" (6351 Republican vs. 414 Democratic speeches) and "Need Fossil Fuels" (4391 Republican vs. 448 Democratic speeches). While claims challenging the fundamental science of climate change, such as "Not Happening," "Not Us," and "Not Bad," remain relatively rare among both parties, Republicans still dominate these categories by margins of 40:1 or greater. The geographic analysis in the bottom panel reveals distinct patterns in the intensity of contrarian discourse when normalized by the number of members of Congress per state and party. For Republican "Solutions Won't Work" arguments, Alaska emerges as the most intensive source with 27.0 speeches per member of Congress, followed by Wyoming (23.6 speeches per member) and West Virginia (22.6 speeches per member). Similarly, for Republican "Need Fossil Fuels" claims, Alaska again leads with an exceptionally high intensity of 40.2 speeches per member, followed by Wyoming (25.3 speeches per member) and West Virginia (22.2 speeches per member). Democratic contrarian claims, while far less numerous overall, show their highest intensities in energy-producing states: West Virginia leads both claim types with 6.0 speeches per member

**Table 1 | Integration of the discourses of delay[3] into the updated CARDS taxonomy**

| Discourses of delay | Updated CARDS taxonomy | Label |
| --- | --- | --- |
| Individualism | Individuals are responsible | 4.2.9 |
| Whataboutism | Other countries' emissions | 4.2.4 |
| Free rider excuse | Policy harms competitiveness | 4.1.1.1 |
| Technological optimism | Technology will fix it | 4.2.10.2 |
| All talk, little action | Already taking it seriously | 4.4.1 |
| Fossil Fuel solutionism | We need fossil fuels | 7.0.0 |
| No sticks, just carrots | Markets are more efficient | 4.2.8 |
| Policy Perfectionism | We're not ready for policy | 4.3.1 |
| Appeal to well-being | Policy harms vulnerable members of society | 4.1.1.2 |
| Appeal to social justice | Policy harms vulnerable members of society | 4.1.1.2 |
| Doomism | It's too late to fix it | 4.3.2 |
| Change is impossible | Solving climate change is too difficult | 4.3.0 |

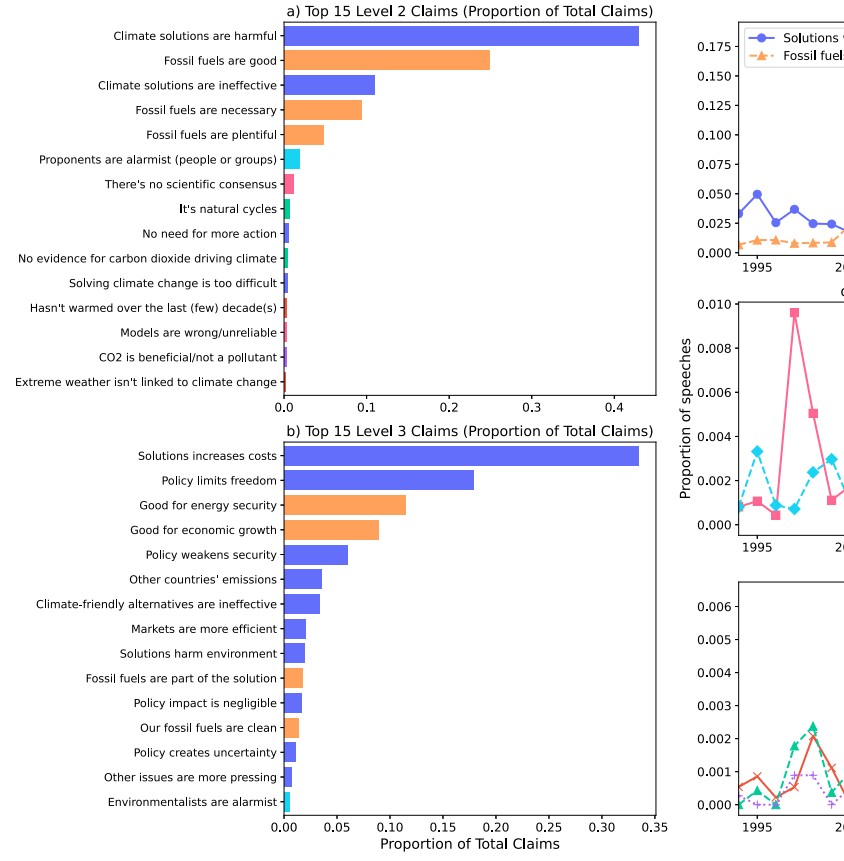

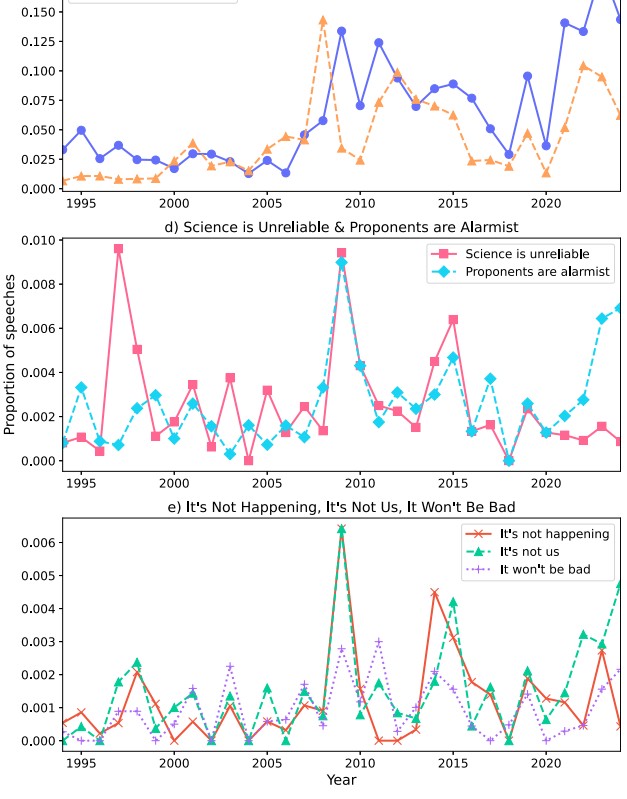

**Fig. 2 | Prevalence of claims.** Figure **a** provides the distribution of the top 15 level-2 claims based on the taxonomy in Fig. 1, while **b** provides the distribution for level-3 claims. Figures **c–e** plot the proportion of each major, level-1 claim from 1994 to

2024. Note that due to major differences in the prevalence of solutions, energy, and science-based claims, the y-axis for Figures **c–e** use different scales.

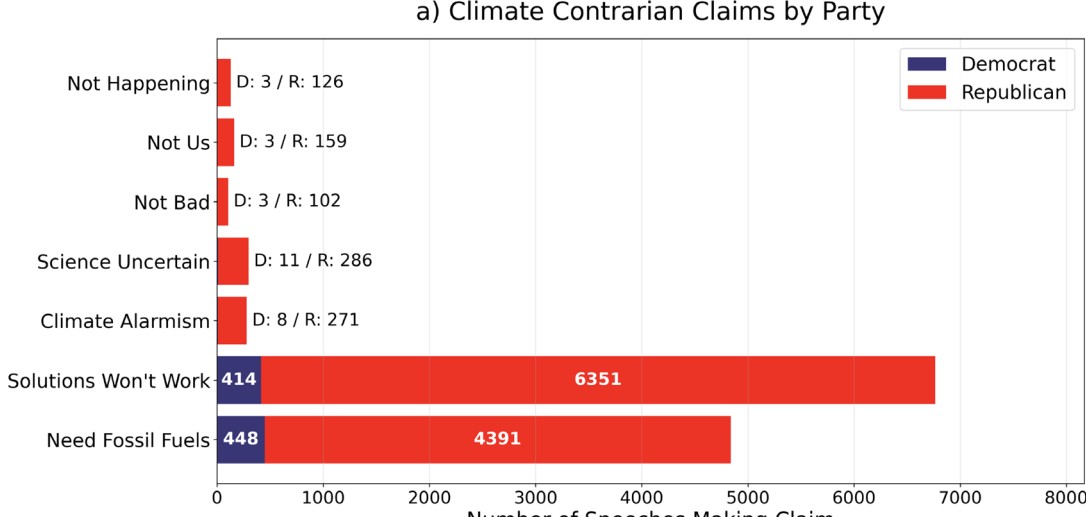

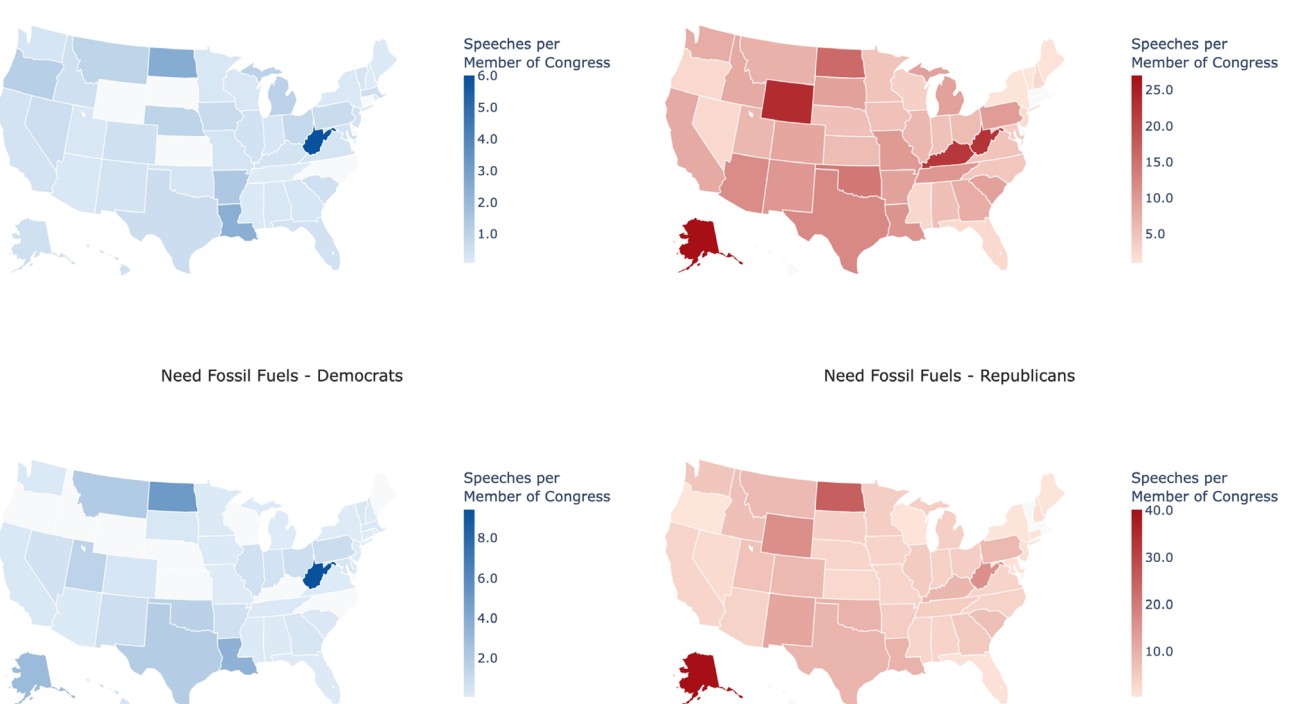

**Fig. 3 | Partisan differences in climate contrarian claims and their geographic distribution across US Congressional speeches, 1994–2023. a** Displays the total number of speeches containing each type of contrarian claim by party affiliation. **b** Illustrates the geographic intensity of "Solutions Won't Work" and "Need Fossil Fuels" claims, showing speeches per member of Congress (Representatives + Senators) for each state and party combination.

for "Solutions Won't Work" and 9.4 speeches per member for "Need Fossil Fuels" arguments. The pattern continues with other fossil fuel-dependent states, as North Dakota ranks second for both claim types (2.5 and 5.0 speeches per member, respectively), followed by Louisiana (2.4 and 3.6 speeches per member). Notably, traditional energy states like Alaska, Montana, Texas, and Oklahoma also appear among the top Democratic sources for "Need Fossil Fuels" claims, while states like Arkansas and Michigan, with significant automotive and manufacturing interests, feature prominently in "Solutions Won't Work" arguments.

**Correlates of contrarian claims making in congressional speech**
Next, we offer a statistical examination of the relationship between several key MoC- and district-level covariates and the likelihood of making a contrarian claim in floor speeches related to climate change. We utilize speech-level data in this analysis, with our primary dependent variable equal to 1 if *any* contrarian claim appears in a floor speech and 0 otherwise. Following previous scholarship on climate change and congressional speech[15], we examine the influence of MoC demographics (sex and age), political variables (party, ideology, chamber, and the party of the President)

**Table 2 | Comparison of full and republican-only models showing the log-odds with 95% credible intervals**

| Variable | All parties | | Republicans | |
|---|---|---|---|---|
| | Estimate | 95% CI | Estimate | 95% CI |
| Intercept | −4.855 | [−5.281, −4.444] | −3.800 | [−4.300, −3.316] |
| Republican | 2.604 | [2.427, 2.789] | – | – |
| Independent | −0.730 | [−2.659, 0.978] | – | – |
| Senate | −0.333 | [−0.495, −0.164] | −0.372 | [−0.536, −0.206] |
| Female | −0.557 | [−0.804, −0.314] | 0.008 | [−0.265, 0.279] |
| Age | −0.160 | [−0.229, −0.091] | −0.111 | [−0.182, −0.041] |
| Fossil fuel contributions | 0.067 | [0.023, 0.113] | 0.044 | [−0.005, 0.093] |
| Fossil fuel employment | 0.240 | [0.178, 0.300] | 0.141 | [0.081, 0.200] |
| Democrat in power | 0.458 | [−0.02, 0.965] | 0.488 | [−0.075, 1.052] |
| Ideology | – | – | 1.472 | [1.254, 1.697] |

Note that Table S2 in Section S.2 of the Supplementary Information demonstrates that these effects are robust to variation in the number and length of floor speeches, which we control for using the total number of words spoken in floor speeches (on any topic) for each MoC in our dataset by year.

and campaign contributions from fossil fuels companies, and district-level data on percentage of employment in fossil fuel industries (see Section in Methods and Section S.3 in the Supplementary Information for a detailed description of covariates). We employ a Bayesian mixed effects logistic regression model given the binary nature of our dependent variable (see Section Statistical methods for more information).

Table 2 provides an overview of our statistical results. Turning first to the "All parties" model, unsurprisingly, we find that political party is by far the strongest predictor of contrarian speech: Republicans are ~12 times more likely to mention a contrarian claim in floor speeches compared to Democrats, with predicted probabilities of 9.5% and 0.77%, respectively. We also find that female MoCs, older MoCs, and senators were less likely to make contrarian claims. Members receiving higher levels of fossil fuel campaign contributions and representing districts (or states for the Senate) with higher levels of fossil fuel-related employment are more likely to make a claim. However, the substantive effects for these variables are relatively weak. For instance, moving from the mean to two standard deviations above the mean for fossil fuel employment only leads to a rise in the predicted probability of making a claim from 0.77% to 1.24%.

As demonstrated in Fig. 2a, Republicans are responsible for the vast majority of the likely obstructionist claims made in floor speeches and thus we provide a second set of estimates to examine *within* Republican party variation. For the "Republicans" model in Table 2, we add MoC left-right ideology (i.e., their DW-NOMINATE score) to examine the hypothesis that more conservative Republicans have a higher likelihood of making contrarian claims. We find that moving from the mean DW-NOMINATE score for Republicans to one standard deviation above the mean raises the likelihood of making a contrarian claim from the baseline of 2.2 to 8.9%. Further, for the Republicans model, age and fossil fuel employment maintain significance based on their 95% credible intervals.

## Discussion

Our analysis reveals several important patterns in climate contrarian discourse within the U.S. Congress that both confirm and extend existing scholarship on climate obstruction. Most notably, "solutions increase costs" comprises nearly 34% of all contrarian claims. Frequent arguments about the necessity of fossil fuels align with Lamb et al.'s[3] identification of appeals to well-being, social justice, and free rider excuses as increasingly prominent discourses of delay in climate obstruction. These findings suggest the validity of both schemes and demonstrate the value of understanding the connection between broad discourses and specific claims. The temporal patterns we observe-with contrarian claims spiking during key legislative moments such as the Waxman-Markey bill in 2009 and the Paris Agreement in 2015-demonstrate how obstructionist discourse in Congress

responds strategically to policy windows, consistent with prior work on the role of elite cues in shaping climate politics[17,18]. And while science-related claims were less prominent than claims about solutions, attacks on climate science also spiked around politically important events, with challenges to the scientific consensus peaking during the Kyoto Protocol period (1997), and then again during cap-and-trade negotiations (2008-9).

The stark partisan divide documented in Fig. 2a, with Republicans responsible for 93% of contrarian claims across all categories, reinforces existing evidence of climate change as a polarizing issue in American politics[30]. However, our geographic analysis in Fig. 2b provides new insights into the spatial intensity of climate contrarianism when controlling for delegation size. The analysis reveals that Alaska, Wyoming, and West Virginia (small-delegation, resource-rich states) exhibit the most intensive climate contrarian rhetoric per member of Congress, with Alaska leading both major claim categories (27.0 speeches per member for "Solutions Won't Work" and 40.2 speeches per member for "Need Fossil Fuels"). The finding that Republican contrarian intensity is almost 12 times higher than Democratic intensity for "Solutions Won't Work" claims and 6.8 times higher for "Need Fossil Fuels" claims shows not just partisan differences in volume but in the systematic deployment of opposition rhetoric. For Democrats, coal-producing West Virginia emerges as the clear outlier, leading both claim categories with intensities that far exceed other Democratic delegations. This finding suggests that local economic interests can override partisan tendencies in fossil-fuel dependent constituencies. The statistical analysis supports the role of fossil fuel campaign contributions and employment in predicting contrarian speech, though the relatively modest effect sizes highlight that partisan identity remains the dominant predictor. Even after controlling for important demographic and economic factors, Republicans are much more likely to make contrarian claims in floor speeches, and this effect becomes even stronger for more right-leaning Republicans.

Our methodological contributions also have important implications for future research on climate obstruction. We demonstrated how a small, carefully selected corpus of examples and RECoT fine-tuning may be used to measure contrarian discourse in Congressional speech. Developing a flexible and efficient procedure for fine-tuning large language models capable of classifying detailed contrarian claims at scale represents a significant advance over previous computational approaches, enabling researchers to track specific narrative strategies rather than broad thematic categories. Further, this flexibility is particularly valuable given the volume of content requiring analysis and the documented spread of climate contrarian arguments rapidly in social media environments and beyond their North American origins[11,31]. However, while our computational *approach* is generally applicable to a range of tasks, the *specific model* developed in this study

## Table 3 | Model performance comparison

| Model | F1 | Precision | Recall | Accuracy | Hamming loss | MCC |
|---|---|---|---|---|---|---|
| Zero-shot learning | | | | | | |
| Claude-Sonnet-3.5 | 0.834 | 0.838 | 0.836 | 0.789 | 0.006 | 0.800 |
| Claude-Sonnet-3.7 | **0.881** | **0.890** | **0.879** | **0.836** | **0.005** | **0.846** |
| GPT-4o | 0.671 | 0.681 | 0.670 | 0.628 | 0.010 | 0.637 |
| GPT-4o-Mini | 0.506 | 0.518 | 0.503 | 0.470 | 0.012 | 0.515 |
| Few-shot learning | | | | | | |
| Claude-Sonnet-3.5 | 0.820 | 0.825 | 0.823 | 0.765 | 0.006 | 0.782 |
| GPT-4o | 0.633 | 0.637 | 0.637 | 0.582 | 0.012 | 0.592 |
| GPT-4o-Mini | 0.550 | 0.558 | 0.554 | 0.495 | 0.014 | 0.501 |
| Fine-tuned models | | | | | | |
| CARDS-mini-GPT | 0.815 | 0.828 | 0.811 | 0.772 | 0.006 | 0.773 |
| CARDS-mini-GPT-2024-12-05 | 0.840 | 0.855 | 0.836 | 0.797 | 0.006 | 0.801 |
| CARDS-mini-Sonnet | 0.825 | 0.838 | 0.821 | 0.787 | 0.006 | 0.784 |
| CARDS-mini-Sonnet-2024-12-05 | 0.852 | 0.866 | 0.848 | 0.809 | **0.005** | 0.809 |

A pairwise version of MCC was used, given the multi-label structure of the data. Best performance values are highlighted in bold.
Due to class imbalance, sample-based metrics were used for F1, precision, and recall.

has only been validated against human annotations of Congressional speech. Understanding the carrying capacity of our model to different domains—e.g., social media posts or different country contexts—will require additional benchmarking data and further model testing. It is also clear from the results provided in Table 3 that the model predictions are not perfect and some statements will be mistakenly classified as contrarian (or not contrarian). More research is needed to understand the extent to which these errors are systematic and steps that might be taken to further improve model performance. We view both establishing diverse benchmarks and improving model performance as important areas of future research.

Our findings also underscore the importance of context in distinguishing between legitimate policy concerns and systematic obstruction. As outlined earlier, when it comes to determining the presence of misinformation, there is often a tradeoff for current models of climate contrarianism, especially for short-form text (e.g., social media posts or transcript paragraphs). Either one focuses on *explicit* statements of misinformation—ignoring a large proportion of relevant claims due to the way in which climate misinformation manifests in real-world data—or one classifies relevant claims without assessing their veracity—thereby running the risk of overstating misinformation for texts lacking sufficient context. We adopted the latter approach and so relying on our model as a tool for countering climate misinformation still requires a human in the loop (i.e., the approach is AI-*assisted* as opposed to fully automated). However, this is not to say that the aggregate data generated by our model does not offer insight into systematic obstruction and climate misinformation. Our results demonstrate the extent to which MoCs routinely—and strategically—deploy potentially misleading claims on science and policy to build a narrative of climate obstruction. Even so, these estimates at best provide an upper bound on the presence of climate misinformation in Congress.

Despite ongoing challenges, further integration of psychological research on misinformation with recent advances in LLMs could provide a way forward. Research demonstrates that identifying the presence of misleading rhetorical techniques and reasoning fallacies in contrarian claims offers one method for addressing claims that are challenging for fact-checkers[24,25], and connects discourse analysis to psychological research into countering misinformation through logic-based corrections[9,32]. Further, several recent attempts to develop LLM-based architectures demonstrate the promise and possible pitfalls of building fully automated AI systems to debunk climate misinformation[28,29]. Negotiating this terrain remains a challenge relevant for both automated classification systems and

policymakers seeking to address climate misinformation while maintaining space for democratic debate about policy approaches. This is an important area for future research on climate change obstruction.

## Methods

### Data

This section outlines our corpus on congressional floor speeches, the annotation procedure used to create the test set for assessing model performance from these data, and the covariates used in the statistical analysis. Section S.3 of the Supplementary Information provides detailed information on the data utilized in this study, including the source, variable descriptions, and descriptive statistics for all variables.

**Congressional speeches**. We utilize a publicly available Congressional Record scraper and parser[33] to extract structured data from HTML files that contain the text of the Congressional Record[34]. This scraped data includes transcripts of all speeches delivered on the House and Senate floor by Congress Members from 1994—i.e., when the Congressional Record was first digitized—to 2024, as well as each speech's date and the speaker's bioguide ID. Each speech was divided into paragraphs (or utterances), resulting in a total of 2,515,806 paragraphs over the sample period. Next, to identify relevant paragraphs for our subsequent analysis, we utilize the ClimateBERT model[35] to classify climate and non-climate change paragraphs (see Section S.4 in the Supplementary Information for details). The process resulted in 110,837 relevant paragraphs over the sample period.

**Test set annotation**. To assess our model performance against a human gold standard, we manually annotated a random sample of 2151 paragraphs from the floor speech dataset using the revised CARDS taxonomy. Given that Democrats are far more likely to discuss climate change in floor speeches, we stratify by party and slightly over-sample Republican speeches. We trained a total of 12 annotators, half of whom contributed substantially to the labeling process. All annotators are or have been research assistants in the Climate and Development Lab at Brown University and have taken a semester-long class on climate obstruction. Additionally, all annotators reviewed the previous CARDS taxonomy and associated video, as well as participated in a lecture and subsequent discussion on the taxonomy updates before starting the labeling process. Each annotator is given one random instance from the climate-relevant

paragraphs to label at a time, and each instance is labeled by at least 3 coders. The labeling occurred between October 2023 and February 2024.

As previously noted in Coan et al.[21], this is a challenging annotation task even for experts in climate change skepticism. Unsurprisingly, the agreement among the student coders was low—with an overall (pairwise) agreement of 79.8%, marginal reliability based on Krippendorff's alpha ($\alpha = 0.501$)—and varied considerably across pairs of coders. To mitigate the challenge of accurately assessing model performance in the face of noisy label data, we had an expert in climate change skepticism and misinformation review instances where there was disagreement among the coders ($N = 510$ instances) and provided the final label for that instance. Although not completely error-free, this procedure greatly reduced label noise and helped to ensure a more accurate estimate of model performance.

**Covariates.** Building on past scholarship on interest group influence in Congress[36,37] and on the correlates of congressional speech on climate change[15], we collected data on a range of covariates at the MoC level. First, we draw on metadata obtained from Congress.gov[38] to extract information on the chamber, term, party, age, and state of the MoC associated with each speech. Second, to measure ideology, we use the first dimension of DW-NOMINATE scores provided by Lewis et al.[39], which represents legislators' positions on the dominant ideological dimension in American politics, primarily reflecting liberal-conservative differences on economic and social issues. This dimension explains the largest share of variance in congressional voting behavior and ranges from −1 (most liberal) to +1 (most conservative). Third, we collected data on campaign contributions from the oil and gas industry and the coal mining industry received by the associated MoC of the given speech during the Congress term of the speech. The campaign contribution data comprises contributions from Political Action Committees and individuals associated with each of the three industries. The data was obtained from OpenSecrets's bulk data[40], who derive the data from Federal Election Commission data. The OpenSecrets industry codes we included as part of the fossil fuel industry were: energy production and distribution (E1000), oil and gas (E1100), major (multinational) oil and gas producers (E1110), independent oil and gas producers (E1120), natural gas transmission and distribution (E1140), oilfield service, equipment and exploration (E1150), Petroleum refining and marketing (E1160), gasoline service stations (E1170), fuel oil dealers (E1180), LPG/liquid propane dealers and producers (E1190), and coal mining (E1210). We matched the OpenSecrets bulk data to our congressional speech dataset by matching each member's Bioguide ID to their corresponding CID (opensecrets_id) in the OpenSecrets data using an existing key[41]. Finally, we collected district level employment data to capture the economic importance of fossil fuel industries within each member's constituency from the Bureau of Labor Statistics' Quarterly Census of Employment and Wages (QCEW). Specifically, we obtained annual employment data for industries classified under specific NAICS codes associated with fossil fuel extraction and related activities: oil and gas extraction (211), coal mining (2121), drilling oil and gas wells (213111), support activities for oil and gas operations (213112), support activities for coal mining (213113), fossil fuel power generation (221112), natural gas distribution (2212), and pipeline transportation (486). The county-level employment data was spatially merged with congressional district boundaries using geographic intersection methods, with employment statistics allocated to districts based on the proportion of county area within each district. We normalize these employment figures by total employment for each constituency (district or state depending on House or Senate member). For more information on the procedure used to build the fossil fuel employment data, see Section S.3 of the Supplementary Information.

### Classifying specific contrarian claims

We build on and extend the machine learning framework developed in Coan et al., drawing on recent advances in large language models (LLMs) to overcome several shortcomings of the original CARDS model. First, as made clear in the original Coan et al. study, their model was only appropriate for categorizing contrarian claims *within* a set of known climate skeptics (e.g., Conservative Think Tanks and Skeptical Blogs). The original model is thus less appropriate for classifying claims when source position is unknown and may produce false positives when classifying non-skeptical content[22]. Second, the original model relies on multi-class, not multi-label data—i.e., the model assumes the presence of a single category for each paragraph under consideration. This decision proved challenging when dealing with paragraphs containing multiple claims, which is the rule, not the exception. For example, consider the following excerpt:

> "And the tragedy is if we were allowed to produce, if this Congress would stop locking up the Outer Continental Shelf, if they would open up the reserves in the Midwest which some of them are taking off in the energy bill, we could have adequate natural gas in this country; the price could be affordable; Americans could be warm; and, the very best jobs in America like petrochemical and polymers and plastic and fertilizer and glass and steel plants and bricks could be made in America, and middle-class working Americans could continue to have the jobs that have historically allowed them to live a quality of life and raise their families."

The claim that Congress should "stop locking up the Outer Continental Shelf" suggests that fossil fuels are plentiful and should be used (category 7.1.0), while the statement "we could have adequate natural gas in this country" suggests that fossil fuels are necessary to meet energy demand (category 7.3.0). The excerpt also makes claims related to the economy and jobs. Statements that the "the very best jobs in America...could be made in America" as the result of expanding fossil fuel supply suggests that fossil fuels are important for economic growth and development (category 7.2.1), while the text also suggests that current energy bill undermines the ability for "middle-class working Americans [to] continue to have the jobs that have historically allowed them to live a quality of life" implies that policies related to climate change could kill the "very best jobs" (category 4.1.1.2).

Lastly, though the taxonomy developed in Coan et al. outlines examples of detailed claims challenging climate science and policy solutions, the machine learning model only classifies texts based on 27 level-2 claims. Yet, classifying claims at a granular level is essential for developing an effective response to climate misinformation[24], while detailed classifications related to climate solutions allow researchers to more clearly connect specific claims to the general discourses of delay proposed in Lamb et al.'s study[3]. We address these limitations by developing a multi-label LLM-based framework that classifies texts down to the lowest level of the CARDS taxonomy.

**In-context learning with foundation models.** The concept of in-context learning (ICL) has emerged as a central paradigm for task adaptation in LLMs[42]. At the core of this paradigm is the model's capacity to adapt its behavior based on provided examples, rather than having its internal parameters systematically modified via resource-intensive fine-tuning, which is a notable shift away from traditional machine learning approaches in NLP. ICL leverages the "context" embedded within the model's prompt to adapt the LLM to specific downstream tasks, spanning a spectrum from zero-shot learning (where no additional examples are provided) to few-shot learning (where several examples are offered). In essence, LLMs are able to execute an array of tasks by conditioning them on a few examples (few-shot) or task-descriptive instructions (zero-shot). This method of conditioning or "prompting" the LLM can be performed either manually[43] or automatically[44].

**Zero-shot classification.** To assess arguably the best case scenario for LLM-based classification performance using ICL, we start by utilizing two frontier models: OpenAI's GPT4o model and Anthropic's Claude Sonnet 3.5 model. As of this writing, these models provide near state-of-the-art (SOTA) performance across a range of benchmarks and thus offer a useful

baseline for understanding the zero-shot capabilities of large foundation models for classifying contrarian claims.

To utilize an LLM-based approach, we must start by specifying a suitable prompt. We follow current best-practices in the prompt engineering literature[45] to iteratively develop the base prompt used in this study (see Section S.5 of the Supplementary Information for the final prompt and a discussion of the prompt development). Our experiments suggest that the use of chain-of-thought (CoT) prompting is particularly helpful for prompt development and classification using foundation models. Zhou et al.[46] carried out a systematic evaluation of different chain-of-thought prompt triggers, including human-designed and Automatic Prompt Engineer (APE). In their experiments, allowing the model to break a problem into steps on their own has proven to be more effective than other prompt triggers. For this paper, we utilized the following APE prompt trigger suggested by Zhou et al.[46]: "Let's work this out in a step by step way to be sure we have the right answer."

**Dynamic few-shot learning**. Research demonstrates that providing examples can allow LLMs to better understand the correlation between the question and the response, thereby improving model performance[47]. Traditional approaches to few-shot learning augment zero-shot prompts by including a handful of examples for each class, often relying on domain experts to hand-pick specific examples or by randomly selecting from a pool of annotated data. There are practical challenges, however, to implementing the traditional approach, especially for an extensive taxonomy such as CARDS. Providing examples of each claim in Fig. 1 would significantly increase the size of the prompt needed for classification, which will increase the cost and latency of model implementation.

Dynamic few-shot learning extends the traditional few-shot approach by only sending examples that are semantically similar to a user's input text. This approach utilizes retrieval augmented generation (RAG) to 1) *retrieve* relevant examples based on calculating the cosine similarity between input text and 2) *generate* few-shot classifications using the selected examples. Dynamic few-shot learning is particularly useful when one has access to a diverse set of training examples—i.e., either sourced through previous research or by hiring annotators—and may provide the only practical approach to implementing few-shot learning for large, multi-label classification problems.

**Fine-tuning scalable alternatives**. Although frontier models and in-context learning have been shown to offer competitive performance across a range of classification tasks, these models are expensive to use for large text datasets, thereby making these models difficult to scale in practice (see Section S.7 of the Supplementary Information for a detailed breakdown of costs). Given challenges associated with scaling current SOTA models, we developed a procedure for fine-tuning a smaller, more scalable (i.e., cheaper and faster) alternative.

**Fine-tuning data**. We start by curating a dataset of high-quality examples for each claim in the revised CARDS taxonomy, aiming for roughly 5 examples per claim. To source examples, we draw on data from the original Coan et al.[21] study, focusing on paragraph level data from conservative think tanks (CTTs). When selecting examples, the research team prioritized those that a) effectively captured the claim of interest and b) provided a diverse representation of the ways in which claims are made in real-world texts. We included a carefully selected set of "No claim" examples, again drawing on data from Coan et al.[21]. Specifically, we select a set of semantically and substantively similar texts that express the opposite view of each claim in the taxonomy. Many of these examples were sourced from quotations and references made to mainstream arguments in CTT text. This procedure resulted in 1691 examples representing the full range of claims in the taxonomy. Note that in addition to using these data for fine-tuning, we also use these examples when carrying out dynamic few-shot learning.

Given that the focus of our analysis is congressional speech, we held out an additional 100 examples from the 2151 manually annotated test set paragraphs to provide additional context on the textual characteristics commonly found in congressional testimonies. OpenAI documentation

suggests that we should observe improvements from fine-tuning on 50 to 100 examples, depending on the use case[48]. Considering the potential difficulties associated with our classification task, we start with the upper-bound of this recommendation. As such, this leaves a total of 2051 out-of-sample paragraphs to assess model performance.

**Reverse engineered chain-of-thought prompting**. While large foundation models can apply APE straightaway, smaller models must be taught to "reason". We developed a procedure—reverse engineered chain-of-thought prompting (RECoT)—that uses examples to solve the dual objectives of teaching a scalable model to "reason", while also training the model to identify specific contrarian claims on climate change. RECoT is carried out in two steps. First, we draw on our fine-tuning dataset and a SOTA model to analyze example text and "reason" through to the final answer provided by the researcher. We explored both Claude 3.5 Sonnet and GPT4o for this step, as these were high performing and affordable options at the time of this analysis. Importantly, the model is instructed to "act" as if it has not been given the correct answers. Second, we then fine-tune a smaller, more scalable base model on the entire CoT responses. After assessing several candidate open- and closed-source base models for RECoT fine-tuning, we found GPT4o-mini provided an effective model for this task, as this model is large enough to efficiently learn from our fine-tuning dataset, but also relatively cheap and fast to ensure model scalability. Supplementary Section S.6 provides a detailed explanation of how we employed the RECoT procedure.

**Model performance**. Table 3 compares the overall performance of the various modeling approaches described above, classifying claims down to level 3 of the taxonomy (see Fig. 1). Several notable findings emerge from this analysis. First, and somewhat surprisingly, few-shot learning underperforms relative to its zero-shot counterparts. This finding suggests that the procedures used to select suitable context or examples hinder rather than help classification performance. This result is particularly striking given that we use the same examples for fine-tuning with substantially better results.

Second, zero-shot classification with Claude-Sonnet-3.7 (one of the most advanced frontier models at the time of this analysis) delivers remarkable overall performance, consistently achieving the highest scores across our full range of metrics. While this performance is promising, the inference costs for Claude-Sonnet-3.7 are prohibitively high, making this model difficult—if not impossible—to scale to even moderate-sized classification tasks. As demonstrated in Supplementary Section S.7, using Claude-Sonnet-3.7 is nearly 20 times more expensive than GPT-4o-mini and 10 times more expensive than our fine-tuned alternatives.

In light of these scalability constraints, we turn to examining our various fine-tuned alternatives. We took the original fine-tuning dataset with 1691 examples curated from CTTs and generated chain-of-thought responses using both GPT-4o and Claude Sonnet-3.5, creating two distinct datasets. Using GPT-4o-Mini as our base model, we fine-tuned two models —one with each dataset—which we term CARDS-mini-GPT and CARDS-mini-Sonnet. These models demonstrated substantial performance improvements compared to the base model (GPT-4o-Mini): CARDS-mini-GPT achieved a 30.9 percentage point increase in F1-score relative to the 4o-mini baseline, while CARDS-mini-Sonnet achieved a 31.9 percentage point increase. Moreover, incorporating the 100 congressional testimony-specific training examples further enhanced these metrics. The best performing CARDS model (CARDS-mini-Sonnet-2024-12-05) achieved performance metrics that closely approximate (or equal, in the case of Hamming Loss) those of the Claude-Sonnet-3.7 alternative. Crucially, the inference cost is approximately one-tenth the price, making CARDS-mini-Sonnet-2024-12-05 a viable alternative for classifying large volumes of text data.

## Statistical methods

To examine the relationship between contrarian claims and key demographic, political, and economic covariates in Congressional speech, we estimated a series of Bayesian mixed effects models. Our unit of analysis is

the speech, and the dependent variable of interest is a binary measure for the presence (1) or absence (0) of a contrarian claim in a given speech. While there are several ways to operationalize contrarian speech in our data (each with benefits and drawbacks), our approach captures the adage that it only takes a little dirt to muddy the waters.

Specifically, we examine the correlates of climate change claims using a Bayesian logistic regression model with random effects for legislators and years. Let $y_{it}$ represent whether member of Congress $i$ made a climate change claim during year $t$. We assume the following probability model:

$$y_{it} \sim \text{Bernoulli}(p_{it}) \tag{1}$$

where $p_{it}$ represents the probability that a member of Congress makes a contrarian claim. To model this probability as a function of covariates, we use the logit link function:

$$\text{logit}(p_{it}) = \log\left(\frac{p_{it}}{1 - p_{it}}\right) = \beta_0 + \mathbf{X}it\boldsymbol{\beta} + \alpha_i + \gamma_t \tag{2}$$

where $\beta_0$ is the global intercept, $\mathbf{X}it$ is a vector of covariates for member $i$ in year $t$, $\boldsymbol{\beta}$ is a vector of fixed effect coefficients, $\alpha_i$ is a member-specific random intercept, and $\gamma_t$ is a year-specific random intercept. The covariates include party affiliation (Republican, Independent), chamber (Senate), sex (Female), standardized age, standardized fossil fuel employment, standardized utility sector presence, and standardized fossil fuel campaign contributions. We estimate the model using Bayesian inference with the following priors:

$$y_{it} \sim \text{Bernoulli}(p_{it}) \qquad \text{(likelihood)} \tag{3}$$

$$\text{logit}(p_{it}) = \beta_0 + \mathbf{X}it\boldsymbol{\beta} + \alpha_i + \gamma_t \tag{4}$$

$$\beta_0, \boldsymbol{\beta} \sim \text{Student} - t(4, 0, 2.5) \qquad \text{(priors)} \tag{5}$$

$$\alpha_i \sim \text{Normal}(0, \sigma_\alpha) \tag{6}$$

$$\gamma_t \sim \text{Normal}(0, \sigma_\gamma) \tag{7}$$

$$\sigma_\alpha \sim \text{Normal}^+(0, 1) \tag{8}$$

$$\sigma_\gamma \sim \text{Normal } t^+(0, .5) \tag{9}$$

where Student-$t^+$ denotes the half Student's $t$ distribution constrained to be positive. We use weakly informative priors for all parameters, with Student-$t$ distributions for the fixed effects and half-Student-$t$ distributions for the random effect standard deviations. The model implemented in Stan (http://mc-stan.org) using Hamiltonian Monte Carlo with 4 chains, each with 3000 iterations (including 1000 warm-up iterations).

## Data availability
All data used in this study is available from Figshare at https://doi.org/10.6084/m9.figshare.30666263.

## Code availability
All the code and prompts developed as part of this study are available at https://github.com/project-c3ds/cards-2pO-paper.

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

## Acknowledgements

The authors thank Ty Pham-Swann, Ava Ward, Xiaokang Xue, and Ethan Drake for research assistance in formulating the project and hand labeling Congressional data. T.C. was supported by the Economic and Social Research Council [grant numbers UKRI147 and ES/W00805X/1].

## Author contributions

The paper's authorship contributions are as follows: T.C., R.M., T.R., and J.C. led on the conceptualization and design of the study; M.N., T.C., and R.M. led on taxonomy changes and development; data collection was carried out by W.K. and C.B.; R.M. and T.C. led on the model design, development, and fine-tuning; T.R. and W.K. organized the human annotation of the test dataset; C.B. and T.C. implemented the data visualizations; T.C. led the statistical analysis; All authors contributed to the analysis and interpretation of the results and reviewed the manuscript.

## Competing interests

The authors declare no competing interests.
