## [Transparent Peer Review file · Communications Sustainability]

Large language model reveals an increase in climate contrarian speech in the United States Congress

Corresponding Author: Professor Travis Coan

Version 0:

Decision Letter:

Dear Professor Coan,

Your manuscript titled "Decoding Delay: Three Decades of Climate Change Opposition in the United States Congress" has now been seen by 3 reviewers, and we include their comments at the end of this message. They find your work of interest, but some important points are raised. We are interested in the possibility of publishing your study in Communications Sustainability, but would like to consider your responses to these concerns and assess a revised manuscript before we make a final decision on publication.

We therefore invite you to revise and resubmit your manuscript, along with a point-by-point response that takes into account the points raised. Please highlight all changes in the manuscript text file.

Please submit your point-by-point responses as a separate file, distinct from your cover letter where you can add responses to the Editors' comments that you do not want to be made available to the reviewers. Word files are preferred. We recommend that any figures, tables or graphs that are included in the response to reviewers are also included in the main article or Supplementary Information.

Please use the following link to submit your revised manuscript, point-by-point response to the reviewers' comments (which should be in a separate document to any cover letter), a tracked-changes version of the manuscript (as a PDF file) and the completed checklist:

Link Redacted

We hope to receive your revised paper within six weeks; please let us know if you aren't able to submit it within this time so that we can discuss how best to proceed. If we don't hear from you, and the revision process takes significantly longer, we may close your file. In this event, we will still be happy to reconsider your paper at a later date, as long as nothing similar has been accepted for publication at Communications Sustainability or published elsewhere in the meantime.

Please do not hesitate to contact us if you have any questions or would like to discuss these revisions further. We look forward to seeing the revised manuscript and thank you for the opportunity to review your work.

Best regards,

Chenchen Ren, PhD
Editorial Board Member
Communications Sustainability

orcid.org/0000-0003-3308-2447

Yann Benetreau, PhD
Consulting Editor, Communications Earth & Environment
Deputy Editor, Communications Sustainability
Nature Portfolio
NY office

EDITORIAL POLICIES AND FORMATTING

- Behavioural and social science
- Ecological, evolutionary & environmental sciences
- Life sciences

Furthermore, please align your manuscript with our format requirements, which are summarized on the following checklist: <https://www.nature.com/documents/commsj-phys-style-formatting-checklist-article.pdf> Communications Sustainability formatting checklist

and also in our style and formatting guide <https://www.nature.com/documents/commsj-phys-style-formatting-guide-accept.pdf> Communications Sustainability formatting guide .

***** DATA:** Communications Sustainability endorses the principles of the Enabling FAIR data project (<http://www.copdess.org/enabling-fair-data-project/>). We ask authors to make the data that support their conclusions available in permanent, publicly accessible data repositories. (Please contact the editor if you are unable to make your data available).

All Communications Sustainability manuscripts must include a section titled "Data Availability" at the end of the Methods section or main text (if no Methods). More information on this policy, is available at <http://www.nature.com/authors/policies/data/data-availability-statements-data-citations.pdf>.

MACHINE LEARNING: For articles that use machine learning techniques, please include a completed version of the ML-reporting-summary. You can download the form here: <https://www.nature.com/documents/ML-reporting-summary.pdf>

If a community resource is unavailable, data can be submitted to generalist repositories such as <https://figshare.com/> or <http://datadryad.org/> Dryad Digital Repository. Please provide a unique identifier for the data (for example a DOI or a permanent URL) in the data availability statement, if possible. If the repository does not provide identifiers, we encourage authors to supply the search terms that will return the data. For data that have been obtained from publicly available sources, please provide a URL and the specific data product name in the data availability statement. Data with a DOI should be further cited in the methods reference section.

REVIEWER COMMENTS:

Reviewer #1 (Remarks to the Author):

This is an innovative, rigorous study leveraging machine learning to better understand opposition to climate action in the U.S. Congress. As far as I can tell, the data and methods are sound. The core results are not particularly surprising, and there don't seem to be major policy or political implications. That said, given the quality of the work, I would support publication without major revision.

Reviewer #2 (Remarks to the Author):

This manuscript presents a comprehensive analysis of “contrarian” climate change discourse in the U.S. Congress from 1994–2023. Using a revised and expanded version of the CARDS taxonomy and a fine-tuned large language model, the authors classify over 110,000 climate-related congressional floor speech paragraphs to identify specific contrarian claims. They find that arguments about the economic costs of climate solutions and the necessity of fossil fuels dominate opposition discourse, with Republicans making such claims at vastly higher rates than Democrats. The paper also links the prevalence of these claims to political, demographic, and economic factors, showing that party affiliation and ideology are the strongest predictors. Methodologically, the work advances computational detection of climate opposition by introducing a scalable, high-performing multi-label classifier. I really liked the tie-in to Lamb’s discourses of delay work.

This study is in many ways an improvement over previous research and an important extension of Coan et al’s original CARDS study. As the authors note, that study was designed to categorize “contrarian claims within a set of known climate skeptics.” They then go on to say that this new LLM-based method reduces false positives when source position is unknown. That makes sense, and I don’t doubt that this is a big improvement over previous approaches. Still, if I understand the authors’ approach correctly, this model is based entirely on congressional speeches. That is certainly ok, and quite interesting, but the language of the paper makes it seem like this is a more generalizable approach. Perhaps it is, but there is no validation of that in the paper. This is already quite an undertaking, so I am not asking for additional analysis. However, I would like the authors to address the generalizability of their approach. To some extent this would be speculation, but the authors are already alluding to the fact that the US has “exported” climate denialism. Would this approach be able to be applied to the speech of European MPs? To the European public? To the American public? Is it only meant to be applied to legislative discourse?

The findings indicate that legislators from California, Pennsylvania, and Texas are the most likely to utter climate skeptical speech acts. That might be because they are fossil fuel producing states, but they are also large states with lots of representatives. Therefore, those states have more opportunities to be climate deniers. I would like to see those figures normalized by number of representatives. That should be easy enough. There is a related issue that also involves thinking about the number of opportunities for climate skepticism. I assume that there is significant variation in the number and length of floor speeches. This would be more labor intensive, but is there some way to account for that as well? Maybe climate skeptics are also in general long-winded (or vice versa)?

At the authors note it is a challenge to distinguish contrarian claims that aren’t necessarily false from true misinformation. I’d like to see a bit more discussion of exactly how their new method does this.

Finally, a point about terminology; “contrarian,” “opposition,” and “skeptical” are all used, but I’m not sure that the authors mean them to be interchangeable. If possible, it would be good to standardize the terminology, or be explicit about the nuanced differences between those terms.

I think this study is a sufficient enough contribution that it should be published.

Reviewer #3 (Remarks to the Author):

In my view this is a really ambitious and well put together study and it does make a proper contribution to the pretty large body of work on climate contrarianism and discourses of delay. The main argument as I read it is basically two things: first that congressional climate opposition has moved over the years from straight up denial to a more subtle delay rhetoric that leans heavily on talking about costs and whether climate solutions are actually doable, and second that party affiliation, esp being a Republican and having a conservative bent, is the biggest factor in predicting this kind of opposition. They also contend that these trends can be tracked with precision using their improved CARDS taxonomy and the fine tuned LLM classifier they built.

What’s new here I think is twofold. One is the way they’ve applied a really detailed, claims-level taxonomy across congressional speeches over three decades, and the other is how they’ve plugged in machine learning to spot these claims at scale. Lots of earlier work has looked at contrarian talk in Congress and online, but the mix of these elements, and the focus on the fine-grained claims rather than broad categories, feels like a proper advance and a nice addition to the field. I’d say it’s going to be interesting to people in climate politics and political comms, and probably the computational social science crowd as well.

I found the analysis itself quite convincing. The stats modelling is set up sensibly and the write-up is clear. They handle the effect sizes with the right amount of caution too, noting that fossil fuel jobs and contributions matter but not anywhere near as

much as party ID. It also looks like the work can be reproduced fairly easily, with the data and code clearly linked up and a decent amount of methodological detail included.

My quibbles are pretty small. First, while the refinement of the taxonomy makes sense, some readers will probably want more on those edge cases where a policy critique might be genuine and not really obstructionist. They do point out that not every claim is misinformation, but it might be worth unpacking a bit more how to handle legit cost–benefit concerns without just sticking them in the contrarian bucket.

Second, I think it wouldn't hurt to say a bit more about potential biases in using an LLM classifier like this. The metrics look strong, but a nod to possible systematic mis-classification (like maybe the model over-tagging certain rhetoric from Republicans just because of the style) could head off some obvious questions from the computational methods people.

All up though I think it's a rigorous and carefully put together piece of work, and it does a nice job of showing that congressional climate opposition has shifted toward delay over denial, while also offering a toolkit other researchers could run with.

** Visit Nature Portfolio's author and reviewers' website at www.nature.com/authors for information about policies, services and author benefits**

Communications Sustainability is committed to improving transparency in authorship. As part of our efforts in this direction, we are now requesting that all authors identified as 'corresponding author' create and link their Open Researcher and Contributor Identifier (ORCID) with their account on the Manuscript Tracking System prior to acceptance. ORCID helps the scientific community achieve unambiguous attribution of all scholarly contributions. You can create and link your ORCID from the home page of the Manuscript Tracking System by clicking on 'Modify my Springer Nature account' and following the instructions in the link below. Please also inform all co-authors that they can add their ORCIDs to their accounts and that they must do so prior to acceptance.

Version 1:

Decision Letter:

<*** REMEMBER TO ATTACH REVISIONS CHECKLIST (WORD)***>

Dear Professor Coan,

Your manuscript titled "Decoding Delay: Three Decades of Climate Change Obstruction in the United States Congress" has now been seen by our reviewers, whose comments appear below. In light of their advice we are delighted to say that we are happy, in principle, to publish a suitably revised version in Communications Sustainability.

We therefore invite you to revise your paper one last time to address the remaining concerns of our reviewers. At the same time we ask that you edit your manuscript to comply with our format requirements and to maximise the accessibility and therefore the impact of your work.

EDITORIAL REQUESTS:

*****Please take care to match our formatting and policy requirements. We will check revised manuscript and return manuscripts that do not comply. Such requests will lead to delays. *****

SUBMISSION INFORMATION:

OPEN ACCESS:

Communications Sustainability is a fully open access journal. Articles are made freely accessible on publication. For further information about article processing charges, open access funding, and advice and support from Nature Portfolio, please visit <https://www.nature.com/commssustain/open-access>

Link Redacted

Best regards,

Chenchen Ren, PhD
Editorial Board Member
Communications Sustainability
orcid.org/0000-0003-3308-2447

Yann Benetreau, PhD
Deputy Editor, Communications Sustainability
Consulting Editor, Communications Earth & Environment
Nature Portfolio
NY office

REVIEWERS' COMMENTS:

Reviewer #1 (Remarks to the Author):

I support this strong manuscript's publication as is.

Reviewer #2 (Remarks to the Author):

I appreciate the authors' careful attention to my suggestions and those of the other reviewer. I support publication in its current form. The only thing I'll add here is related to my comment 2.3. I do think it would be good to add their analysis of the total words spoken variable to the supplementary appendix even if the results are largely consistent with the initial analysis.

Reviewer #3 (Remarks to the Author):

Thanks again for the chance to look over the revised version of this piece. I've now read through both the updated manuscript and the responses, and I think the revisions have indeed strengthened the paper from what was already a really solid piece. The authors have done a good job of responding to the feedback, and the result is a clearer and sharper piece of work. In my view, the paper is ready for publication.

** Visit Nature Portfolio's author and reviewers' website at <http://www.nature.com/authors> for information about policies, services and author benefits**

Reviewer 1

Comment 1.1: This is an innovative, rigorous study leveraging machine learning to better understand opposition to climate action in the U.S. Congress. As far as I can tell, the data and methods are sound. The core results are not particularly surprising, and there don't seem to be major policy or political implications. That said, given the quality of the work, I would support publication without major revision.

Response 1.1: Thank you for your comments and supporting our research. We agree that the statistical results are not particularly surprising, but instead work to reinforce existing work in the field (albeit with a more comprehensive dataset).

Reviewer 2

Comment 2.1: Still, if I understand the authors' approach correctly, this model is based entirely on congressional speeches. That is certainly ok, and quite interesting, but the language of the paper makes it seem like this is a more generalizable approach. Perhaps it is, but there is no validation of that in the paper. This is already quite an undertaking, so I am not asking for additional analysis. However, I would like the authors to address the generalizability of their approach. To some extent this would be speculation, but the authors are already alluding to the fact that the US has "exported" climate denialism. Would this approach be able to be applied to the speech of European MPs? To the European public? To the American public? Is it only meant to be applied to legislative discourse?

Response 2.1: This is an excellent comment and we agree that we needed to be more careful here. While the computational *approach*—i.e., RECoT fine-tuning—is quite general and applicable across a range of classification tasks, the specific *model* described in the paper has only been validated for Congressional speech. We've made several edits throughout the manuscript to provide clarity on the issue of generalizability. Importantly, we add the following additions to the discussion section:

“Our methodological contributions also have important implications for future research on climate obstruction. We demonstrated how a small, carefully selected corpus of examples from and RECoT fine-tuning may be used to measure contrarian discourse in Congressional speech. Developing a flexible and efficient procedure for fine-tuning large language models capable of classifying detailed contrarian claims at scale represents a significant advance over previous computational approaches, enabling researchers to track specific narrative strategies rather than broad thematic categories. Further, this flexibility

is particularly valuable given the volume of content requiring analysis and the documented spread of climate contrarian arguments rapidly in social media environments and beyond their North American origins (Treen et al. 2020; McKie 2021). However, while our computational *approach* is generally applicable to a range of tasks, the *specific model* developed in this study has only been validated against human annotations of Congressional speech. Understanding the carrying capacity of our model to different domains---e.g., social media posts or different country contexts---will require additional benchmarking data and further model testing.”

While speculative at this point, there are several other reasons to believe that the approach developed will generalize better than previous approaches. For instance, the core fine-tuning dataset was produced using text from conservative think tanks (CTTs), while the model was tested on Congress, suggesting at least some capacity to generalize. Moreover, given that the model is built on an LLM, translation-related issues should also be easier to overcome (or at least mitigate). In preliminary work benchmarking the model using old Twitter and in a different language (i.e., Czech), we are seeing very promising results. However, as you remark, this was already quite an undertaking and we need to leave this additional work for future research.

Comment 2.2: The findings indicate that legislators from California, Pennsylvania, and Texas are the most likely to utter climate skeptical speech acts. That might be because they are fossil fuel producing states, but they are also large states with lots of representatives. Therefore, those states have more opportunities to be climate deniers. I would like to see those figures normalized by the number of representatives.

Response 2.2: This is an excellent suggestion and how the results should have been presented the first time. We have updated Figure 3 to present the normalized results and this, as the reviewer suggests, offers a much clearer picture. We also updated relevant text in the Results and Discussion section on the geographic distribution of claims making. For example, here is the relevant text from the Results:

The geographic analysis in the bottom panel reveals distinct patterns in the intensity of contrarian discourse when normalized by the number of members of Congress per state and party. For Republican “Solutions Won’t Work” arguments, Alaska emerges as the most intensive source with 27.0 speeches per member of Congress, followed by Wyoming (23.6 speeches per member) and West Virginia (22.6 speeches per member). Similarly, for Republican “Need Fossil Fuels” claims, Alaska again leads with an exceptionally high intensity of 40.2 speeches per member, followed by Wyoming (25.3 speeches per member)

and West Virginia (22.2 speeches per member). Democratic contrarian claims, while far less numerous overall, show their highest intensities in energy-producing states: West Virginia leads both claim types with 6.0 speeches per member for "Solutions Won't Work" and 9.4 speeches per member for "Need Fossil Fuels" arguments. The pattern continues with other fossil fuel-dependent states, as North Dakota ranks second for both claim types (2.5 and 5.0 speeches per member respectively), followed by Louisiana (2.4 and 3.6 speeches per member). Notably, traditional energy states like Alaska, Montana, Texas, and Oklahoma also appear among the top Democratic sources for "Need Fossil Fuels" claims, while states like Arkansas and Michigan, with significant automotive and manufacturing interests, feature prominently in "Solutions Won't Work" arguments.

It is important to note, however, that the general finding on the prominence of fossil fuel-rich states holds after normalization. Nevertheless, we agree with the reviewer that this offers a more accurate visual representation of spatial variation in claims making.

Comment 2.3: There is a related issue that also involves thinking about the number of opportunities for climate skepticism. I assume that there is significant variation in the number and length of floor speeches. This would be more labor intensive, but is there some way to account for that as well? Maybe climate skeptics are also in general long-winded (or vice versa)?

Response 2.3: It certainly feels like climate skeptics in Congress are more long-winded! We tested the impact of this on our results by going back to the original data and measuring the total number of words spoken in floor speeches (on any topic) for each MoC in our dataset by year. The table below provides the results for adding this measure as an additional covariate in our Full and Republican-only models. As demonstrated in the table, the results for our core variables of interest are nearly identical after controlling for the *Total Words Spoken* variable. If anything, the results are slightly stronger for several indicators when including the total words control variable.

Given that there is virtually no difference between these new results and our original findings, we've opted to keep the original specification, as we do not have theoretical expectations for the *Total Words Spoken* variable. We could, however, put these additional results in an appendix if the reviewer has strong preferences here.

Variable	Full Model		Republican Model	
	Estimate	95% CII	Estimate	95% CI

Intercept	-4.92	[-5.347, -4.498]	-3.846	[-4.349, -3.342]
Republican	2.634	[2.461, 2.812]	—	—
Independent	-0.781	[-2.77, 0.907]	—	—
Senate	-0.287	[-0.458, -0.119]	-0.332	[-0.505, -0.155]
Female	-0.565	[-0.814, -0.328]	0.011	[-0.26, 0.278]
Age	-0.161	[-0.23, -0.091]	-0.11	[-0.182, -0.04]
FF Contributions	0.01	[-0.018, 0.037]	0.007	[-0.022, 0.036]
FF Employment	0.237	[0.174, 0.3]	0.138	[0.076, 0.2]
Democrat in Power	0.469	[-0.033, 0.963]	0.492	[-0.055, 1.036]
Total Words Spoken	-0.077	[-0.144, -0.011]	-0.07	[-0.143, 0.004]
Ideology	—	—	1.485	[1.264, 1.707]

Comment 2.4: At the authors note it is a challenge to distinguish contrarian claims that aren't necessarily false from true misinformation. I'd like to see a bit more discussion of exactly how their new method does this.

Response 2.4: Thanks for this comment. We have tried to make clear in the text that our model simply classifies claims and “any model developed to classify these claims cannot, on its own, detect misinformation.” While many of the claims included in taxonomy are regularly used to support statements of misinformation (as demonstrated in previous literature; e.g., see Cook, Ellerton, Kinkead 2018), their use can be context dependent (a limitation shared by *Discourses of Delay* as well). As we've tried to illustrate through the “climate models are uncertain” example, there is a tradeoff when modelling these types of claims: either a) you focus on when claims *explicitly* use misinformation and potentially miss a large proportion of claims due to the way in which climate misinformation manifests in real-world data OR b) you simply focus on classifying relevant claims without having the model assess the veracity of the claim. In terms of distinguishing actual misinformation, given that the main goal of our model is to create an *AI-assisted* approach to fighting misinformation—i.e., where a human is still in the loop versus a fully automated approach without human intervention—we chose to prioritize the latter approach (b). However, we agree with the reviewer that we needed to be clearer on the implications of this tradeoff.

To help clarify, we've added the following text to the discussion:

“Our findings also underscore the importance of context in distinguishing between legitimate policy concerns and systematic obstruction. As outlined earlier, when it comes to determining the presence of misinformation, there is often a tradeoff for current models of climate contrarianism, especially for short-form text (e.g., social media posts or transcript paragraphs). Either one focuses on *explicit* statements of misinformation---ignoring a large proportion of relevant claims due to the way in which climate misinformation manifests in real-world data---or one classifies relevant claims without assessing their veracity---thereby running the risk of overstating misinformation for texts lacking sufficient context. We adopted the latter approach and so relying on our model as a tool for countering climate misinformation still requires a human in the loop (i.e., the approach is AI-assisted as opposed to fully automated). However, this is not to say that the aggregate data generated by our model does not offer insight into systematic obstruction and climate misinformation. Our results demonstrate the extent to which MoCs routinely---and strategically---deploy potentially misleading claims on science and policy to build a narrative of climate obstruction. Even so, these estimates at best provide an upper bound on the presence of climate misinformation in Congress.”

Here, we reiterate this tradeoff, point out the necessity of human input if the model is used to facilitate fact-checking, and mention why we believe that our approach is valuable for studying systematic obstruction and misinformation. Importantly, while the results for model precision outlined in Table 3 do not suggest a systematic bias towards false positives, we make clear that these estimates at best provide an upper bound on misinformation. We hope that this provides sufficient clarification and thank the reviewer again for this helpful comment.

Comment 2.5: Finally, a point about terminology; “contrarian,” “opposition,” and “skeptical” are all used, but I’m not sure that the authors mean them to be interchangeable. If possible, it would be good to standardize the terminology, or be explicit about the nuanced differences between those terms.

Response 2.5: This is another helpful suggestion and we needed to be more careful here. We have edited the draft to use consistent language throughout, changing the terminology to “contrarian” where appropriate.

Reviewer 3

Comment 3.1: First, while the refinement of the taxonomy makes sense, some readers will probably want more on those edge cases where a policy critique might be genuine

and not really obstructionist. They do point out that not every claim is misinformation, but it might be worth unpacking a bit more how to handle legit cost–benefit concerns without just sticking them in the contrarian bucket.

Response 3.1: This is an important comment and a very similar one was raised by Reviewer 2 (see Comment 2.4 above). We’ve provided additional clarity in the Discussion section on what we see as the main tradeoff when classifying claims and making clear that our results “at best provide an upper bound on the presence of climate misinformation in Congress” (see Response 2.4 above for the full response). Although our estimates for model precision compared to annotators are strong, a certain amount of false positives are unavoidable when trying to also balance the need for recall. Hopefully, the additional clarifying text (see also Response 2.4) is sufficient to address the reviewer’s concern.

Comment 3.2: Second, I think it wouldn’t hurt to say a bit more about potential biases in using an LLM classifier like this. The metrics look strong, but a nod to possible systematic mis-classification (like maybe the model over-tagging certain rhetoric from Republicans just because of the style) could head off some obvious questions from the computational methods people.

Response 3.2: This is another helpful suggestion and builds on points made by other reviewers. As the reviewer suggests, we offer a nod to potential biases, adding the following text to the discussion:

However, while our computational *approach* is generally applicable to a range of tasks, the *specific model* developed in this study has only been validated against human annotations of Congressional speech. Understanding the carrying capacity of our model to different domains---e.g., social media posts or different country contexts---will require additional benchmarking data and further model testing. It is also clear from the results provided in Table 3 that the model predictions are not perfect and some statements will be mistakenly classified as contrarian (or not contrarian). More research is needed to understand the extent to which these errors are systematic and steps that might be taken to further improve model performance. We view both establishing diverse benchmarks and improving model performance as important areas of future research.

Here, we make clear that this over- or under-tagging is taking place, but point out that we are still uncertain about whether these errors are due to systematic features of the textual inputs and how one might improve model performance in the future. Hopefully, this goes some way towards alleviating the reviewer’s concerns.

Review 2

Comment 2.1: I do think it would be good to add their analysis of the total words spoken variable to the supplementary appendix even if the results are largely consistent with the initial analysis.

Response 2.1: Thank you again for all the helpful comments. As per your suggestion we added the robustness check including the Total Words Spoken measure to the Supplementary Material (see Section S.2, Table S.2) and reference the table in main results table in the manuscript (Table 2).